

# An androgen receptor-based signature to predict prognosis and identification of ORC1 as a therapeutical target for prostate adenocarcinoma

Linjin Li[1], Dake Chen[1], Xiang Chen[1], Jianlong Zhu[1], Wenshuo Bao[1], Chengpeng Li[1], Feilong Miao[1] and Rui Feng[2]

[1] Department of Urology, The Third Clinical Institute Affiliated to Wenzhou Medical University, The Third Affiliated Hospital of Shanghai University, Wenzhou People's Hospital, WenZhou, Zhejiang, China

[2] Zhenjiang Hospital of Chinese Traditional and Western Medicine, Zhenjiang, Jiangsu, China

Corresponding author
Rui Feng, 13913439473@139.com

## ABSTRACT

**Background:** Aberrant activation of androgen receptor (AR) signaling plays a crucial role in the progression of prostate adenocarcinoma (PRAD) and contributes significantly to the development of enzalutamide resistance. In this study, we aimed to identify a novel AR-driven signature that can predict prognosis and endows potentially reveal novel therapeutic targets for PRAD.

**Methods:** The Seurat package was used to preprocess the single-cell RNA sequencing (scRNA-seq). Differentially expressed genes were visualized using limma and pheamap packages. LASSO and multi-variate Cox regression models were established using glmnet package. The package "Consensus Cluster Plus" was utilized to perform the consensus clustering analysis. The biological roles of origin recognition complex subunit 1 (ORC1) in PRAD were determined by gain- and loss-of-function studies *in vitro* and *in vivo*.

**Result:** We characterized the scRNA-seq data from GSE99795 and identified 10 AR-associated genes (ARGs). The ARGs model was trained and validated in internal and external cohorts. The ARGs were identified as an independent hazard factor in PRAD and correlated with clinical risk characteristics. In addition, the ARGs were found to be correlated with somatic tumor mutation burden (TMB) levels.

Two groups that have distinct prognostic and molecular features were identified through consensus clustering analysis. ORC1 was identified as a critical target among these ARGs, and it ORC1 promoted proliferation and stem-like properties of PRAD cells. Chromatin immunoprecipitation (ChIP)-qPCR assay confirmed that AR could directly bind the promoter of ORC1. Activated AR/ORC1 axis contributed to enzalutamide resistance, and targeting ORC1 rendered PRAD cells more susceptible to enzalutamide.

**Conclusions:** This study defines an AR-driven signature that AR activates ORC1 expressions to promote PRAD progression and enzalutamide resistance, which may provide novel targets for PRAD treatment.

## INTRODUCTION

Prostate cancer (PCa) is a common malignancy threatening the health and life of men worldwide, which is aggressive to the male genitourinary system (*Zhu et al., 2021*). According to the 2021 cancer statistics, the estimated new PCa cases is up to 248,530, and the cancer-related deaths are about 34,130 in the USA (*Siegel et al., 2021*). Apparently, the incidence and mortality of PCa increased year by year, accounting for more than 90% of reproductive organ cancer diseases (*Zelic et al., 2020*). As is well known, PCa is a highly heterogeneous tumor, and the overall prognosis of PCa patients varies differentially. Prostate adenocarcinoma (PRAD), which expresses androgen receptor (AR), is the most frequent histological subtype of Pca (*Chen, Chu & Lin, 2022*). For patients with early-stage diseases, the combined strategies of PCa were overall effective, including radical prostatectomy, androgen deprivation therapy (ADT), chemotherapy, and radiotherapy (*Gamat & McNeel, 2017*). Nevertheless, approximately 25% of patients would develop into advanced stages with distal metastases and drug resistance. Particularly, the development of a highly aggressive castration-resistant PCa (CRPC) would result in poor survival and limited treatment strategies for patients (*Fan et al., 2021*). As a result, there is a pressing need in developing more effective strategies and identifying potential innovative targets to improve the overall survival of PRAD patients. Owing to the remarkable heterogeneity of PCa, patients with distinct prostate-specific antigen (PSA) levels or Gleason scores may have various prognosis even at the same clinical stages. Therefore, we intended to find more effective diagnostic and prognostic markers for PRAD.

Given that the AR signaling plays an essential role in the progression of PCa, ADT, like bicalutamide and abiraterone, has become the foundation for patients with locally advanced or metastatic PRAD (*Lorente et al., 2015*). However, the lethal CRPC state is a serious problem for patients and there are currently no effective managements (*Hong et al., 2021*). Intensive data have indicated that aberrant activation of AR signaling and downstream targets are essential determinants for the progression of CRPC (*Chandrasekar et al., 2015*). Enzalutamide, as a novel inhibitor of the AR pathway, directly binds to the ligand-binding domain of AR, thus suppressing nuclear translocation and transcriptional activity of AR. Nonetheless, the development of enzalutamide resistance (Enza-R) is an inevitable problem for patients with advanced disease (*Antonarakis et al., 2014*; *Annala et al., 2018*). Significant insights into the mechanisms of Enza-R have been provided, including amplification of AR or AR-related genes, abnormal expression, generation of AR splice variants, and activation of AR bypass crosstalk or downstream targets. Genomic-wide sequencing based on large PCa samples has identified amplification and activation of AR genes, which account for nearly 50% of all CRPC patients. Particularly, various key regulators such as HOXB13, FOXM1, or FOXA1, have been found to recruit AR or its variants to specific promoters, to sustaining cell growth and promoting progression of PRAD even in the presence of enzalutamide (*Ketola et al., 2017*; *Song et al., 2019*). These results indicate that sustained AR expression or activation of downstream targets represents a major vulnerability in the treatment of Enza-R PRAD.

Recently, single-cell genomics, such as single-cell RNA sequencing (scRNA-seq), have emerged as powerful tools for exploring the genetic or functional heterogeneity, reconstructing evolutionary lineages, and identifying rare subpopulations (*Ziegenhain et al., 2017*). Particularly, scRNA-seq has proven effective in transcriptional classification of cell types such as breast cancer, lung adenocarcinoma, and pancreatic cancer (*Kalucka et al., 2020*; *Katzenelenbogen et al., 2020*; *Chen et al., 2021*). Besides, scRNA-seq in human tumors has provided valuable insights into tumor heterogeneity and the existence of distinct subpopulations, which are crucial for understanding cancer-related mechanisms. For instance, a recent study on human lung cancers utilized the scRNA-seq in 49 clinical biopsies identified active T-lymphocytes and decreased macrophages in residual disease and immunosuppressive cells in progressive disease (*Maynard et al., 2020*). Another study conducted the scRNA-seq on docetaxel-sensitive and -resistant variants of PRAD DU145 and PC3 cell lines, revealing NUPR1 as a mediator of PRAD drug resistance (*Schnepp et al., 2020*). Given that scRNA-seq data of PRAD cells or samples are publicly available, we aimed to integrate scRNA-seq data with RNA-seq data deposited in TCGA to identify potential AR-related genes (ARGs) that contribute to progression and drug resistance in PRAD.

In the present study, we performed a re-analysis of scRNA-seq data from PRAD cells obtained from the GSE99795 dataset to explore AR-related drivers. Multiple PRAD cohorts were utilized to identify a series of ARGs with significant prognostic value. Functional experiments were conducted to highlight the role of ORC1 as a crucial factor in PRAD cell proliferation and Enza-R, thereby providing insights for individualized treatment strategies in PRAD.

## METHODS AND MATERIALS

### Acquisition of cell samples and PRAD population cohorts

The GSE99795 dataset containing raw data from 144 PRAD cells treated with androgen at 0 and 12 h was downloaded from the Gene Expression Omnibus (GEO) dataset (https://www.ncbi.nlm.nih.gov/geo/query/acc.cgi?acc=GSE99795). In this project, the researchers conducted transcriptome profiling of 144 single LNCaP PCa cells, both treated and untreated with androgen, following cell cycle synchronization.

For additional data, we downloaded the RNA-seq and clinical information of The Cancer Genome Atlas (TCGA)-PRAD samples *via* Genomic Data Commons (GDC) portal (https://portal.gdc.cancer.gov/). The normalization of transcriptome count was conducted using the edgeR package (Version 3.26.8). The expression data and clinical information of other PRAD samples were obtained from GSE116918, GSE70769, and the MSKCC-PRAD cohort (https://www.cbioportal.org/study/summary?id=prad_pik3r1_msk_2021) for more intensive analysis.

### Processing of single-cell RNA-seq data

We collected scRNA-seq data of 144 tumor cells using the GRCh38 reference genome, which contain three types of cells: 0-h untreated cells, 12-h untreated cells and 12-h androgen-treated cells. We utilized the Seurat package to construct the object and applied
quality control filters to remove cells with poor quality (*Yip, Sham & Wang, 2019*). The scRNA-seq was performed using the 10× Genomics platform and the Illumina HiSeq 2500 sequencing technology. We assessed the percentage of gene counts, cell counts, and mitochondria sequencing counts to ensure data integrity. In addition, we performed principal component analysis (PCA) as a linear dimensionality reduction technique. This allowed us to identify the most significant dimensions of dataset based on an estimated $P$ value.

## Identification of ARGs model in PRAD population cohorts

We extracted the expression data of essential genes from the TGCA-PRAD patients and obtained their corresponding clinical information. The least absolute shrinkage and selection operator (LASSO) regression analysis was conducted by the glmnet package to identify the essential targets. Survival analysis was conducted using the survival package, specifically employing the Kaplan-Meier method. To visualize the differential distributions of the ARGs signature, we utilized bubble plots and scatter graphs. Additionally, the multigene activity (MAG) signature was calculated using the formula: $MAGs = \Sigma (\beta i \, ^* \, Expi)$, where $\beta i$ represents the weight of each identified target. In the training TCGA-PRAD cohort, we performed receiver operating characteristic (ROC) curve to evaluate the predictive significance of ARGs. The difference in progression-free outcomes was assessed through Kaplan-Meier analysis with the log-rank test. Furthermore, the ggDCA package was utilized to conduct the decision curve analysis (DCA).

## Profiles of tumor mutation burden and correlation analysis

The tumor mutation burden (TMB) calculation in the TGCA-PRAD dataset was performed using the following formula: TMB = (total variant count)/(total length of exons). A Perl script was developed to handle the mutational data from PRAD patients, which included deletions, insertions, and substitutions across bases. The Maftools package was utilized to visually represent the mutation profiles of the two risk levels using a waterfall plot. Subsequently, the chi-square test was employed to detect and compare the differential mutation frequencies of mutants between the two groups of ARGs. Additionally, TMB was determined for each sample, and the potential associations between ARGs and TMB levels were assessed through the estimation of $P$ values.

## Consensus clustering

Consensus clustering (*Wu & Liang, 2021*) was conducted using the R package "Consensus Cluster Plus." The optimal number of subgroups was evaluated using the cumulative distribution function (CDF) and consensus matrix. Furthermore, PCA analysis was employed to identify two distinct groups characterized by different molecular processes and prognostic outcomes.

## Functional analysis

Since we have already divided the TCGA-PRAD cohort into two groups with high and low ARGs levels, we further conducted gene set enrichment analysis (GSEA) using the ARGs score as the phenotype. Using the GSEA software and the Java platform, we utilized the

"c2.cp.kegg.v6.2.symbols.gmt gene sets" from the MSigDB database (http://software.broadinstitute.org/gsea/msigdb) as the reference set. The enriched crosstalk with false discovery rate < 0.05 were regarded to be statistically significant.

## Cell lines and cell culture

The 293 T cells and human PRAD cell lines (22Rv1 and C42-B) were acquired from the American Type Culture Collection (ATCC). The 22Rv1 and C42-B cells were cultured in Roswell Park Memorial Institute-1640 medium (Invitrogen, Waltham, MA, USA) supplemented with 10% fetal bovine serum (FBS, HyClone, Logan, UT, USA). Authentication of all cell lines was performed through short tandem repeat (STR) profiling. The cells were maintained in a humidified incubator at 37 °C with 5% $CO_2$. Regular testing for mycoplasma contamination was conducted using the LookOut Mycoplasma PCR Detection Kit (Sigma-Aldrich, St. Louis, Mo, USA).

## Plasmid construction, transfection, and establishment of stable cell lines

The full-length cDNA encoding human ORC1 was amplified through polymerase chain reaction (PCR) and inserted into a pMSCV-puro-retro vector (TaKaRa, Kusatsu, Shiga, Japan) by cloning. Subsequently, two short hairpin RNAs (shRNAs) targeting ORC1 were cloned into the pLKO.1-puro vector (Invitrogen, Waltham, MA, USA). Specific regions of ORC1 promoter sequences were amplified using PCR and cloned into a luciferase reporter plasmid. The transfection of the plasmids into the designated cells was performed using Lipofectamine 3000 (Invitrogen, Waltham, MA, USA). To establish stable cell lines, retroviral or lentiviral infection was employed to express Flag-ORC1 or ORC1-shRNA, respectively. Following a 10-day culture period, the cells were selected using 0.5 µg/mL puromycin. Double-strand oligos of sh-ORC1: 5′-CACCAGTTTTCGATCCAACAAGGGCCGAAGCCCTTGTTGGATCGAAAAC-3′.

## Cell counting kit-8 and 3D soft agar assay

Cell viability was assessed using the cell counting kit-8 (CCK-8) kit (Dojindo Molecular Technologies, Mashikimachi, Japan). Following the transfection of C4-2B and 22RV-1 cells with shORC1 lentivirus for 48 h, the cells were seeded in 96-well plates with 1,000 cells per well and five replicate wells. CCK8 reagent was added to the wells, mixed with the cell culture medium, and incubated for 3 h. The optical density (OD) at 450 nm was measured using a microplate reader (ELX-800; BioTek, Winooski, VT, USA) to generate a growth curve.

For the soft agar assay, cells from each group were trypsinized to obtain a single-cell suspension. After cell counting, the cell concentration was adjusted to $5 \times 10^2$ cells/ml, and 2 ml of the suspension was seeded into each well of a six-well plate. Three parallel wells were set up for each group. The six-well plate was placed in a 37 °C, 5% $CO_2$ incubator for regular culture, with the medium changed every 2 days. On the 14th day, the six-well plate was removed, and the medium was discarded. The plates were rinsed three times with physiological saline, and the cells were fixed with 1 ml of 4% paraformaldehyde for 30 min.

After removing the fixative, each well was stained with 1 ml of 5% crystal violet dye for 1 h. Cell colonies containing ≥50 cells were counted as individual colonies under a microscope.

### Sphere formation assay

The cells were plated in either a six-well plate coated with poly-HEMA (Sigma, Burlington, MA, USA) or an ultra-low-attachment plate (Corning, Corning, NY, USA). They were then incubated in serum-free medium supplemented with 1× N2 supplement, 20 ng/mL bovine fibroblast growth factor, and 20 ng/mL epidermal growth factor for a duration of 10 days. The number of spheres formed was counted under a microscope, specifically focusing on spheres with a diameter greater than 100 μm.

### Animal experiments

Male NOG mice, aged 4 weeks and weighing 20 ± 2.0 g, were utilized as transplant recipients. All procedures involving animals were conducted in accordance with the institutional ethical requirements and were approved by the Institutional Animal Care and Use Committee of Wenzhou Medical University (Approval No. WMU2071357-AC-03). The sample size for the animal experiment was determined as five nude mice in both the shCtrl group and shORC1 group, and the mice were randomly assigned. A cell suspension of approximately $5 \times 10^6$ C42-B cells in PBS (200 microliters) was injected into 4-week-old NOG nude mice, with five mice in each group. Simultaneously, the mice were treated with enzalutamide at a dosage of 2.5 mg/kg/d. Tumor size was measured every 3 days using calipers, and tumor volume was calculated using the formula: V $(mm^3)$ = a × $b^{2/2}$, where "a" and "b" represent the long and short diameters, respectively. At 28 days, the mice were sacrificed, and the subcutaneous tumors were excised and preserved at −80 °C.

### Statistical analysis

The differences between two groups were assessed using the Student's t-test, while the log-rank test was employed to analyze survival differences. All experimental data were presented as mean ± standard deviation (SD). Statistical analysis was performed using GraphPad Prism 7.0 software. A $P$-value less than 0.05 was considered statistically significant.

## RESULTS

### Single-cell RNA-seq profiling and TCGA-PRAD cohort identify a panel of ARGs in PRAD

To identify the novel AR-driven signature in PRAD, we integrated scRNA-seq, TCGA-PRAD and other PRAD datasets to identify the essentially oncogenic AR-drivers. The supplementary assays were also conducted to demonstrate the *in vitro* and *in vivo* roles of identified targets. The screening procedures were illustrated on Fig. 1. To deal with the scRNA-seq data, we downloaded the raw data from the GEO dataset *via* the accession number of GSE99795. In this study, we found that 48 LNCaP single cells and one representative bulk cell RNA sample (1 ng) were collected for SMART-seq2 amplification and later single-cell RNA-seq (total 144 single cells and three bulk cell samples) for each of

three treatment groups. All of the treatment groups were harvested from after synchronizing the cells at the G1/S phase with a double thymidine block and androgen depriving the cells for ~24 h. Treatment groups 2 and 3 were cultured in the absence and presence of androgen (1 nM R1881) for 12 h, respectively. Treatment group 1 was a baseline comparison treatment group and was collected right after cell synchronization and androgen deprivation (considered 0 h). Then, the scRNA-seq data of 171 files was downloaded and combined into one matrix and we transformed the gene symbols by referencing the gene transfer format (GTF) file. We exhibited the quality control chart in Fig. 1A, in which the range of detected gene numbers and the sequencing numbers of each cell lines were shown. In this process, the cells with a percentage of mitochondrial sequencing count >5% were therefore excluded. Meanwhile, the significantly positive associations between the detected gene accounts and the sequencing depth were observed with Pearson's r = 0.25 (Fig. 2B). The cluster analysis exhibited two distinct groups between data derived from the 12 androgen-treated cells and those derived from the 12 h untreated cells (Fig. 2B). In addition, we performed the variance analysis, where the top 10 significantly differentially expressed genes (DEGs) were exhibited and noted in red, like EAF2, CDC20, PLK1, as well as CCNB2 (Fig. 2C). We extracted the top 800 variant genes based on the scRNA-seq and conducted the cluster analysis in three group samples (Control; 12 h−, R1881; 12 h+, R18821), which was illustrated in the heatmap diagram (Fig. 2D). The principal component analysis based on the scRNA-seq could successfully distinguish the groups in either 12 h androgen-treated cells and 12 h untreated cells, indicating the differential transcriptome features within them (Fig. 2E). Gene ontology (GO) analysis suggested that these DEGs were mainly enriched in cell cycle, lipid biosynthetic process, dendrite development, as well as fatty-acyl-CoA metabolic process (Fig. 2F).

## Establishment of ARGs model in TCGA-PRAD cohort

To identify the essential ARGs with prognostic significance in PRAD, we selected the top 53 DEGs and performed the LASSO analysis, where 10 hub genes were found (Figs. 3A and 3B). Accordingly, the significant differential expression of 10 ARGs in two clusters were shown in dot plot (Fig. 3C). The ARGs signature was thus constructed *via* multi-variate Cox regression analysis, where 3-, 5-, and 7-year of the areas under the curve (AUCs) of the ROC curves were 0.789, 0.799 and 0.837, respectively (Fig. 3D). Lastly, we calculated the ARGs scores for each patient in TCGA-PRAD cohort and divided the samples into high-risk and low-risk groups, and Kaplan-Meier analysis revealed that patients in high-risk groups suffered from worse survival outcomes relative to those from the low-risk groups with log-rank test $P < 0.001$ (Fig. 3E).

## Risk assessment and predictive efficacy of ARGs in internal and external PRAD cohorts

Since we have already established the ARGs model, we intended to figure out the underlying relationships between ARGs and other clinical variables. Expectedly, high ARGs scores correlated positively with clinical T stages ($P = 3.414e{-}15$), positive rates of

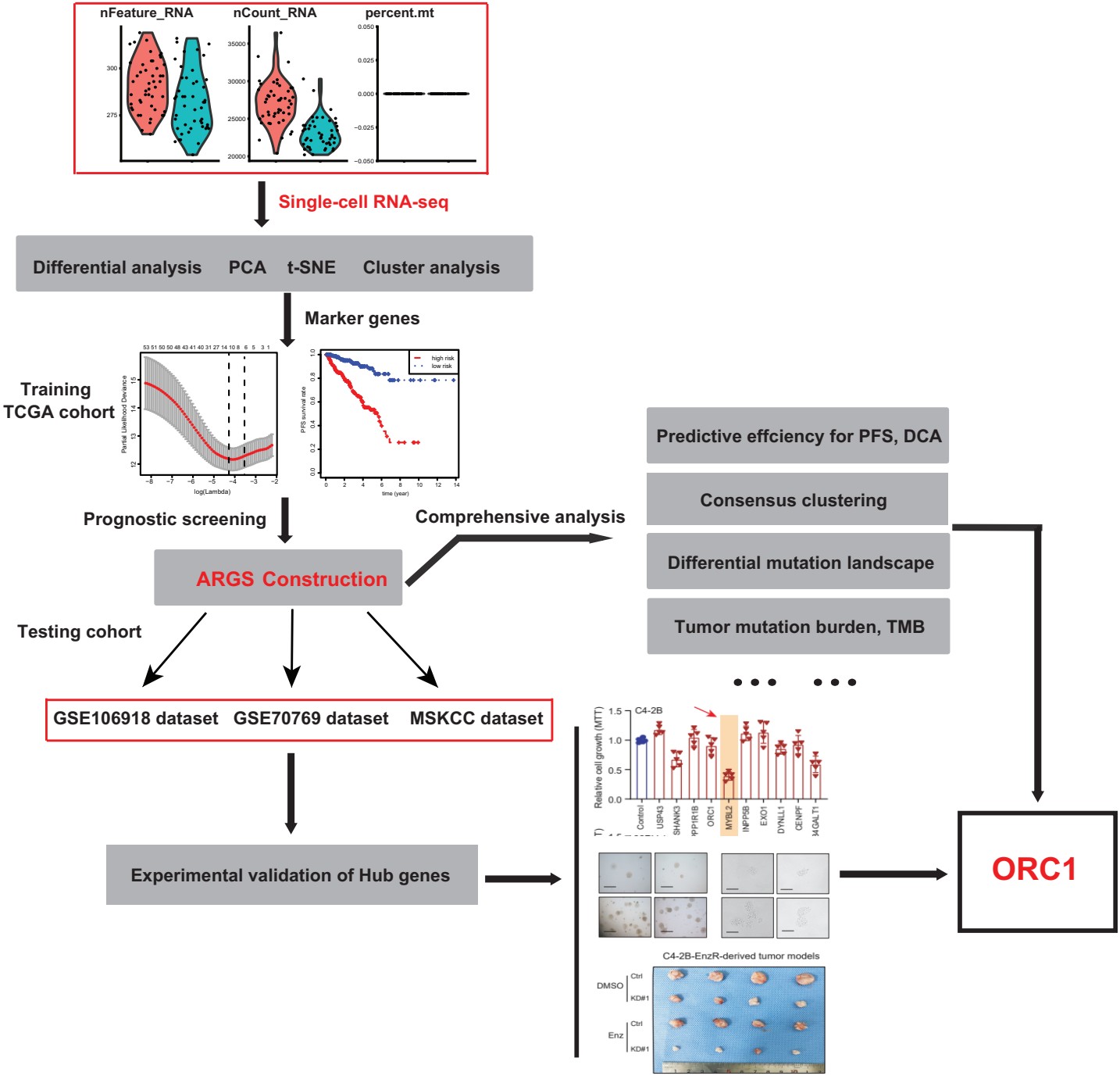

**Figure 1 The schematic flowchart of screening of ARGs signature and identification of ORC1 in PCa.**

lymph nodes ($P$ = 3.815e−08), as well as advanced Gleason scores ($P$ = 3.352e−23) (Figs. 4A–4C). Besides, we integrated the common risk variables in PRAD and performed the uni-variate Cox regression analysis and found four risk factors in PRAD, like Gleason scores, ARGs, T and N stages (Fig. 4D). Subsequently, multi-variate Cox regression analysis further demonstrated that ARGs is an independent factor in PRAD, along with the

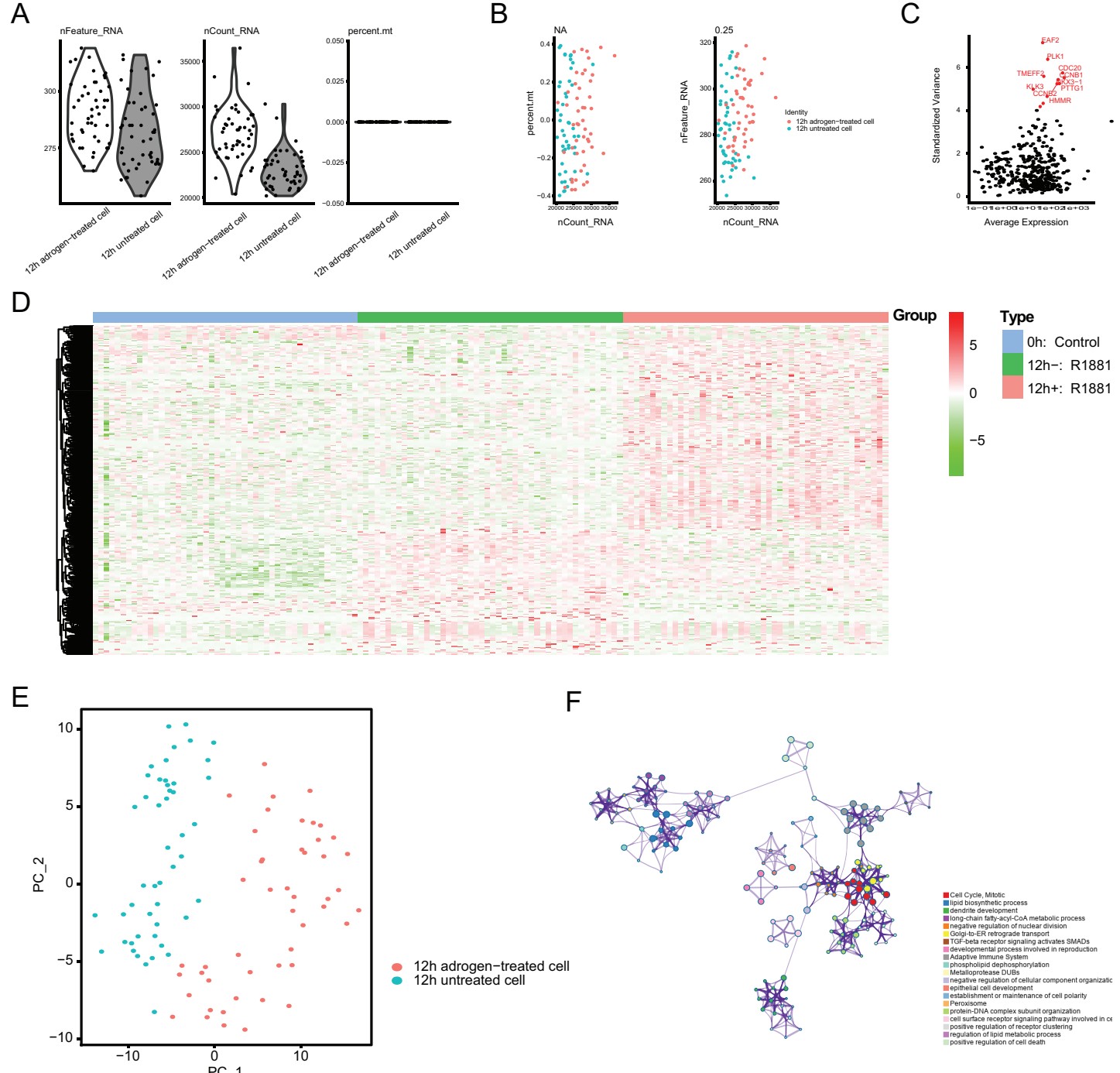

**Figure 2** Single-cell RNA-seq (scRNA-seq) analysis identified top differentially expressed genes associated with androgen treatment. (A and B) Quality control of scRNA-seq for cell sub-populations. The cells with poor quality were filtered out and we analyzed the positive associations between detected gene counts and sequencing depth. (C) The gene symbols with significant difference across cells were noted in red and we illustrated them by the characteristic variance diagram. (D) Cluster analysis of DEGs across three cell groups by heatmap, including 0 h: Control, 12 h−: R1881, 12 h+: R1881. (E) The principal component analysis (PCA), a linear dimensionality reduction method, was used to find the significantly available dimensions of data sets with the estimated *P* value, where we divided the cell groups into two categories. (F) Gene ontology (GO) analysis *via* Metascape platform identified the top significantly enriched crosstalk.

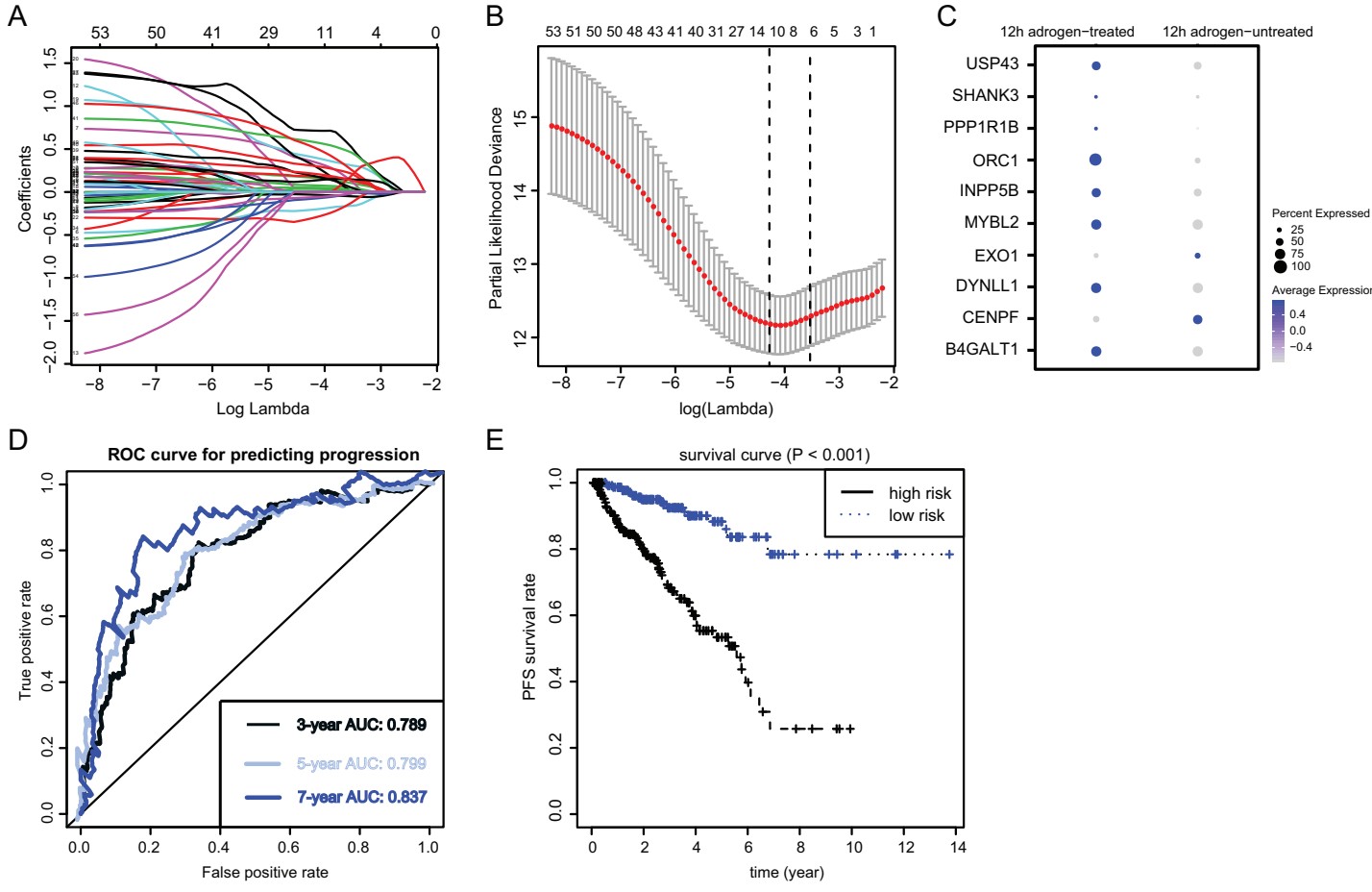

**Figure 3 Identification and construction of ARGs in PCa samples.** (A and B) We conducted the LASSO algorithmto identify the 10 prognostic genes in TCGA training cohort, where the optimal cutoff value was −4 and the minimum account of genes was 18. (C) The bubble plot exhibited the significantly differential expressions of 10 prognostic genes. (D) The receiver operating characteristic curve (ROC) analysis showing the predictive efficiency of 3-, 5- and 7-year in predicting PFS of patients. (E) Kaplan-Meier analysis showing the differential survival outcomes between ARGs-high and ARGs-low samples.           

traditional factor of Gleason score (Fig. 4E). Given that PSA is a classcal predictor for PRAD risks, we found that PSA levels could effectively predict the progression risks of patients with 3-year AUC = 0.719, 5-year AUC = 0.716 and 7-year AUC = 0.765 (Fig. 4E). Patients with high PSA levels had the shorter progression-free survival compared with those with low Gleason scores, as indicated by the Kaplan-Meier analysis with $P < 0.001$ (Fig. 4F). Lastly, the decision curve analysis (DCA) was further utilized to assess the performance of ARGs model and PSA in predicting clinical outcomes of PRAD patients. DCA demonstrated that the combined model (ARGs and PSA levels) showed the best net benefit for 5-year progression-free survival (PFS) relative to either unique factor alone (Fig. 4G).

The ARGs model was further validated by other three independent PRAD cohorts, including the GSE116918 dataset ($N = 248$), GSE70769 dataset ($N = 92$) and MSKCC cohort ($N = 140$). In GSE116918, the AUCs for 3-, 5-, and 7-year progression-free survival predictions for the risk scores were 0.705, 0.766, and 0.76, respectively. Patients in

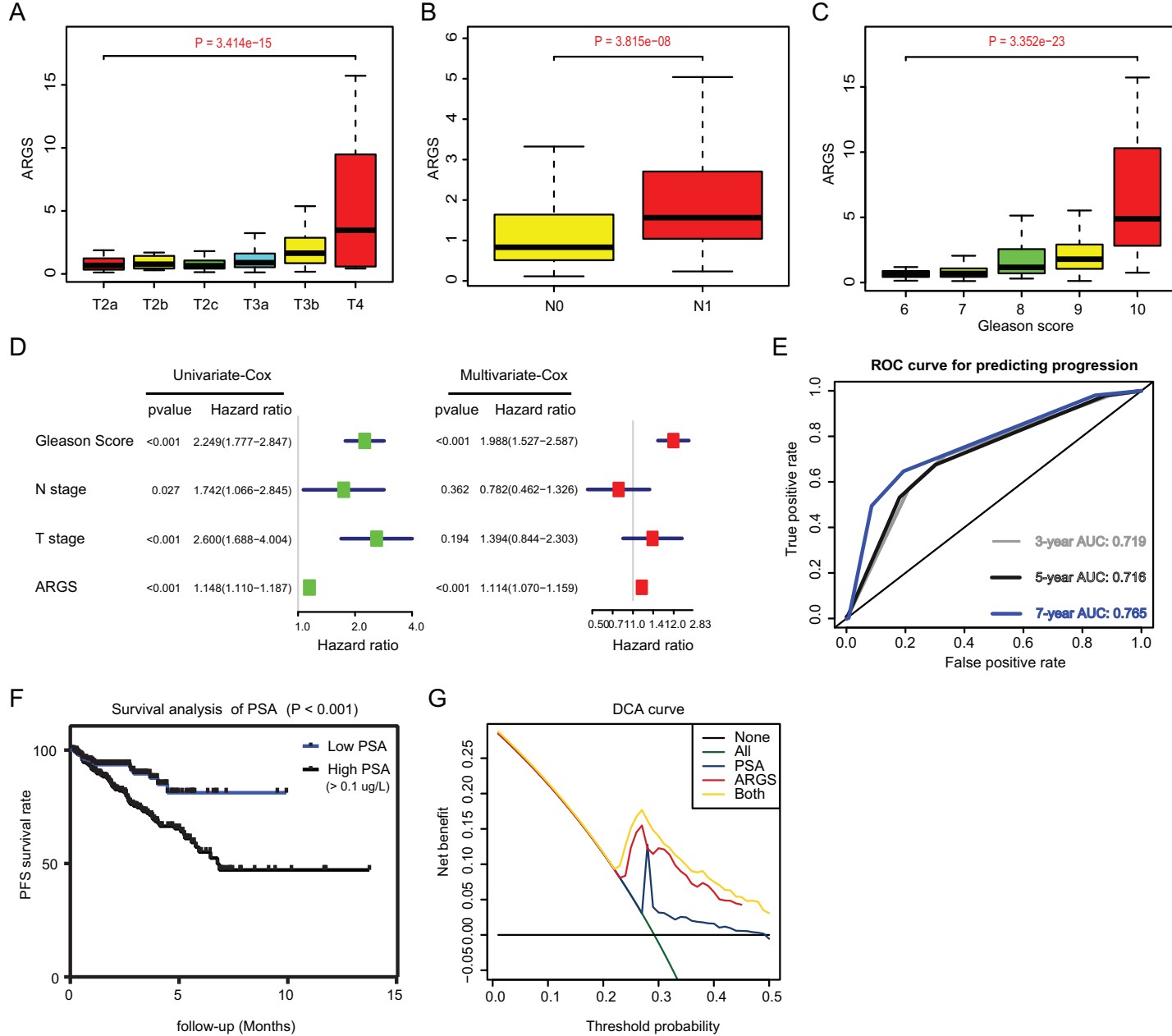

**Figure 4 Correlation analysis between ARGs with clinical factors and prognostic significance of ARGs in PRAD.** (A–C) Kruskal-Wallis test showing that increasing ARGs-score correlated with higher T stages (*P* = 3.414e−15), higher positive rate of lymph nodes (*P* = 3.815e−08) and Gleason scores (*P* = 3.352e−23). (D) Univariate- and multi-variate Cox regression analysis exhibited the robustness of ARGs in PRAD samples. (E) The receiver operating characteristic curve (ROC) analysis showing the predictive efficiency of 3-, 5- and 7-year in predicting PFS of PSA levels. (F) Kaplan-Meier analysis showing the significance of PSA in PRAD samples. (G) DCA analysis was further used to evaluate the performance of ARGs model and PSA levels in predicting clinical outcomes of PRAD patients.

high-risk groups had worse PFS outcomes relative to those in low-risk groups with log-rank test *P* = 6.387e−05 (Figs. 5A and 5B). In addition, in the GSE70769 cohort, the AUCs for 3-, 5-, and 7-year progression-free survival predictions for the risk scores were 0.775, 0.793, and 0.773, individually (Figs. 5C and 5D). Patients in the high-risk groups suffered from the worse PFS outcomes compared with those in low-risk groups with

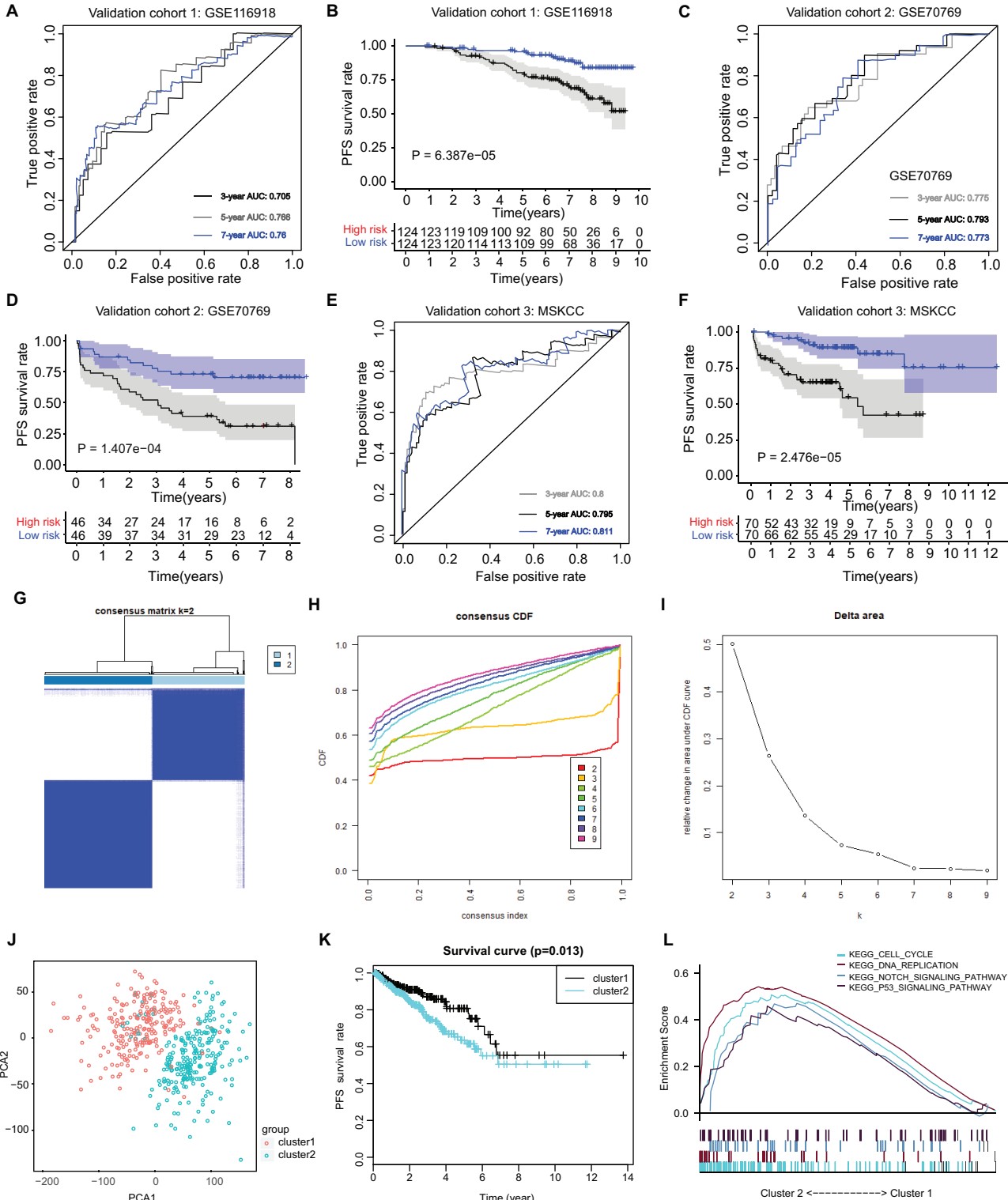

**Figure 5 External validation of ARGs signature and clustering analysis across PRAD samples.** (A) The ROC analysis of ARGs exhibiting its predictive efficiency of 3-, 5- and 7-year in predicting PFS. (B) Kaplan–Meier curves showing the differential survival outcomes in AGRs-high and ARGs-low samples in GSE116918. (C and D) The ROC analysis and Kaplan–Meier curves of ARGs in GSE70769. (E and F) The ROC analysis and Kaplan–Meier curves of ARGs in MSKCC-PRAD cohort. (G) Color-coded heatmap corresponding to the consensus matrix for k = 2 obtained by consensus clustering. The color gradients from 0 to 1 represent the degree of consensus, with white corresponding to 0 and dark blue corresponding
**Figure 5 (continued)**
to 1. (H) Consensus among clusters for each category number k. (I) Delta area curve of consensus clustering indicating the relative change in area under the cumulative distribution function (CDF) curve for each category number k compared to k − 1. (J) PCA analysis indicated the identified two groups based on ARGs signature. (K) Kaplan-Meier analysis indicated the differential survival outcomes of two groups. (L) Gene set enrichment analysis (GSEA) exhibiting the enriched crosstalk between Cluster1 and Cluster2.

log-rank test $P$ = 1.407e−04 (Figs. 5E and 5F). Consistent with the above findings, we observed the similar results in the MSKCC cohort. Collectively, these data suggested that the ARGs model was a robust predictive factor in the PRAD patients.

## Cluster stratification of PRAD samples based on the identified ARGs

Utilizing the unsupervised clustering algorithm, we conducted consensus clustering analysis to distinguish PRAD patients in the training cohort into subgroups based on the expression of ARGs. The K = 2 was identified with the optimal clustering stability (Figs. 5G–5I). We further conducted the PCA analysis and could classify the patients into two distinct groups with individual features (cluster 1 & cluster 2) in Fig. 5J. Kaplan-Meier analysis implicated that patients in the cluster two group had worse PFS outcomes compared with those in the cluster 1 group, in which the log-rank test $P$ value was 0.013 (Fig. 5K). In addition, we also performed the GSEA between the cluster 2 and cluster 1 groups and we found that several oncogenic crosstalk was significantly enriched, including cell cycle, DNA replication, notch signaling pathway and p53 signaling (Fig. 5L).

We further calculated the TMB variable in the TCGA-PRAD cohort, matched with corresponding ARGs scores. The mutational landscape showed that mutation events occurred more frequently in the high-risk ARGs group *versus* low-risk group, as exhibited by the waterfall plot. Particularly, we compared the differential mutation rates of mutated genes distributed in more than 5% of all samples, and TP53, KMT2D or PTEN harbored more mutation sites in high-risk group relative those in the low-risk group, as indicated by the Chi-square test (Fig. S1A). Besides, TMB levels correlated positively with ARGs scores with Pearson's r = 0.278 (Fig. S1B). Intriguingly, TMB is also a prognostic factor in PRAD with log-rank test $P$ = 0.021 (Fig. S1B).

## Experimental validations highlight ORC1 as an essential hit among the ARGs

Given the above success in finding ARGs in PCa, we thus intended to obtain the AR-related regulators that are specifically required for PCa cells. Next, we conducted the validation screens using the individual shRNAs to investigate inhibitory effects in 22RV-1 and C4-2B cells with specific knockdown of identified targets. As shown by the 3-(4, 5-dimethylthiazol-2-yl)-2, 5-diphenyltetrazolium bromide (MTT) assay, we found that silencing of ORC1 showed the most pronounced suppressive effect on cell viability among the candidates including ORC1, USP4S, HANK3, PPP1R1B, MYLB2, INPP5B, EXO1, DYNLL1, CENPF and B4GALT1 (Fig. 6A). ORC1 ablation clearly led to the most decrease of cell growth in both 22RV-1 and C4-2B cells. Therefore, we focused on ORC1 for

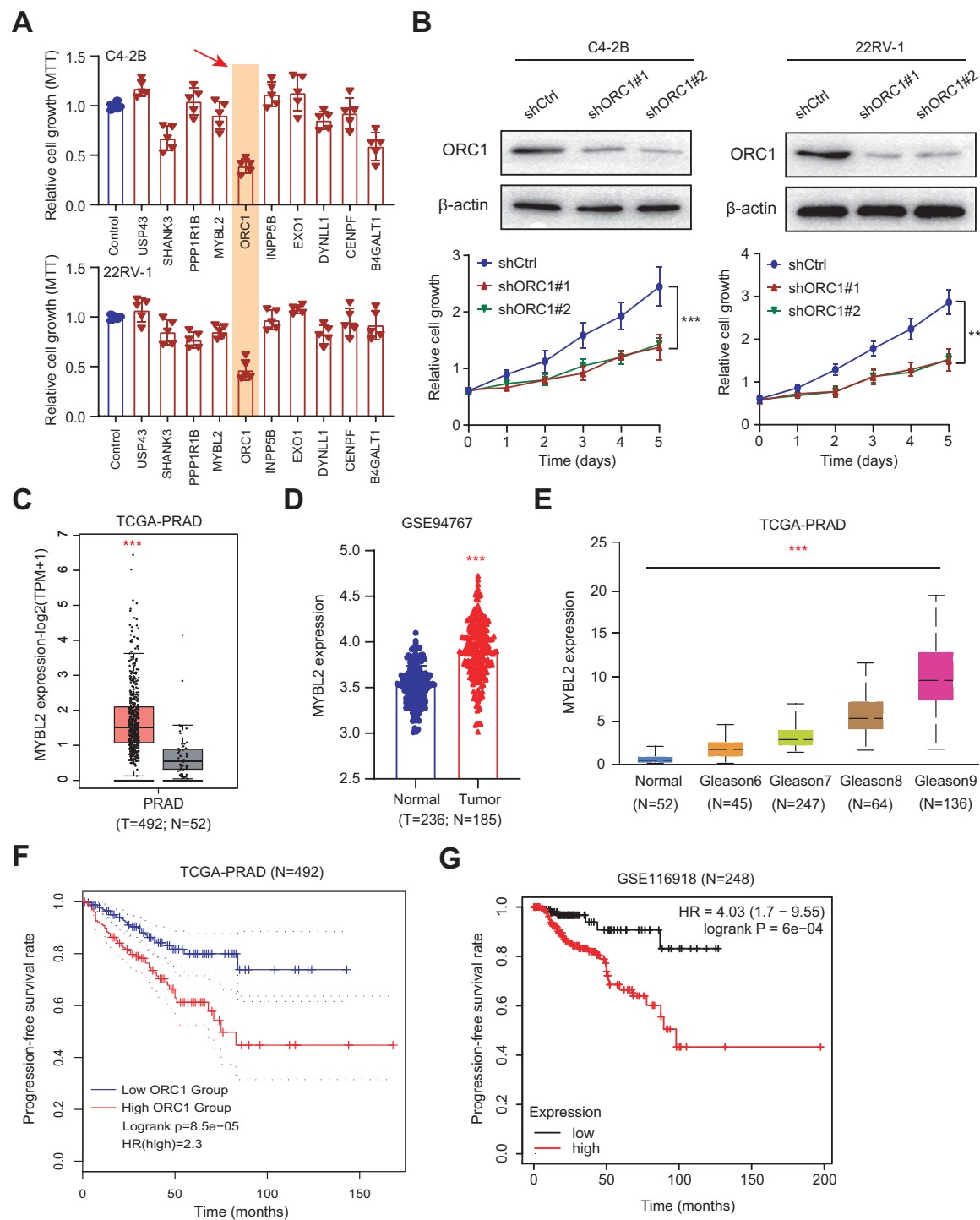

**Figure 6 Experimental validations identified ORC1 as an essential target in PRAD.** (A) MTT assay was used to examine the viability of C4-2B and 22RV-1 after the administration of specific shRNAs of USP4S, HANK3, PPP1R1B, MYLB2, ORC1, INPP5B, EXO1, DYNLL1, CENPF and B4GALT1. (B) CCK8 assays (upper) indicating that ORC1 knockdown remarkably suppressed cell growth in C4-2B and 22RV-1 cells. The knockdown efficiency of ORC1 was confirmed by western blot below. (C) Boxplot showing the differential analysis of ORC1 mRNAs in tumor and normal samples of TCGA-PRAD cohort. (D) Boxplot showing the differential analysis of ORC1 mRNAs in tumor and normal samples in GSE94767. (E) Correlation analysis showing the associations between ORC1 levels and Gleason scores. (F and G) Kaplan-Meier analysis indicating the differential survival outcomes between high-ORC1 and low-ORC1 groups in TCGA-PRAD (F) and GSE116918 (G). ***$p < 0.001$.

following functional studies. Utilizing the shRNAs targeting ORC1, the ORC1 was knocked down in PCa cells, which was confirmed by western blot (Fig. 6B). Expectedly, ORC1 inhibition could notably suppress the cell proliferation rates compared with the cells from control group, as indicated by CCK-8 assays (Fig. 6B). We then detected the expression levels of ORC1 in TCGA-PRAD cohort and observed that ORC1 expressed highly in tumor samples *versus* normal tissues (Fig. 6C). The same results were found in GSE94767 (Fig. 6D). Intriguingly, elevated ORC1 levels correlated with high Gleason score levels (Fig. 6E). Lastly, Kaplan-Meier analysis in TCGA-PRAD and GSE116918 datasets showed that patients with high ORC1 levels had the worse outcomes with shorter PFS months compared with patients with low ORC1 (Figs. 6F and 6G). Taken together, we revealed that ORC1 is an essential AR-related hit in PRAD and has the prognostic significance.

## AR activates ORC1 transcription to drive tumor progression and Enza-R in PRAD cells

To elucidate the function of ORC1 in PRAD, we established stable ORC1-overexpressing PRAD cells (22RV-1 and C4-2B). The up-regulation of ORC1 significantly enhanced PRAD cell soft agar colony formation efficiency (Fig. 7A). Besides, the self-renewal potentiality of PRAD cells was also increased when cells were transfected with ORC1, indicating that ORC1 could also maintain stemness features of PRAD (Fig. 7B). Next, we intended to figure out the underlying regulations between AR and ORC1. Firstly, we found AR overexpression could notably elevate ORC1 expression levels, whereas AR knockdown decreased its levels, along with another known AR downstream target of PSA (Fig. 7C). Positive correlations between AR and ORC1 were confirmed in the TCGA-PRAD samples (Fig. 7D). Additionally, the chromatin immunoprecipitation (ChIP) assay was performed to show the promoter region of ORC1 was enriched in AR binding and H3K27ac signals (Fig. 7E). Knockdown of AR could markedly decrease the enrichment of H3K27ac signals in the promoter of ORC1 (Fig. 7E). These data implicated that AR directly binds at the promoter of ORC1 that accounts for the transcription and upregulation of ORC1. Lastly, in order to determine whether ORC1 regulates Enza-R, we observed that ORC1 overexpression attenuated the sensitivity of C4-2B-Parental cells to enzalutamide, whereas ORC1 ablation compromised the resistance of C4-2B-Enza-R cells to enzalutamide (Fig. 7F). Lastly, we further established the C4-2B-Enza-R tumor xenograft models and found that ORC1 knockdown could render C4-2B-Enza-R tumors sensitive to enzalutamide treatment, as quantified by tumor volumes and weight (Figs. 7G–7I). Collectively, AR could directly activate ORC1 transcription that mediates the Enza-R in PRAD cells.

## DISCUSSION

In efforts to develop more effective therapies, it is critical to identify prognostic factors with high predictive significance and improve the understanding of molecular mechanisms underlying CRPC development (Ge et al., 2020; Zhang et al., 2022). Previous knowledge mainly focused on the screening of biomarkers that expressed differentially between

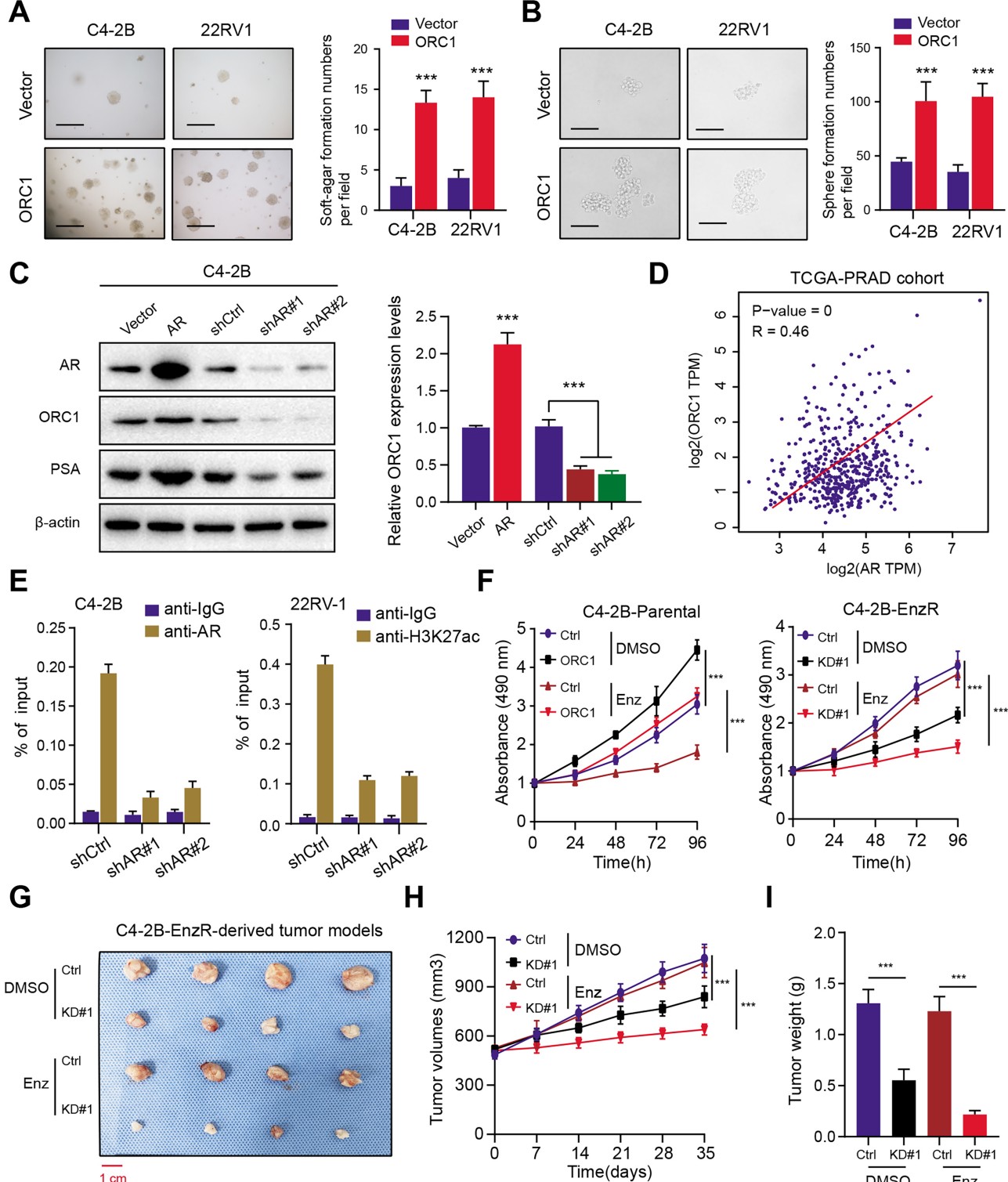

**Figure 7 AR activates ORC1 to sustain tumor progression and enzalutamide resistance in PRAD.** (A) The anchorage-independent growth of C4-2B and 22RV-1 cells in soft agar (scale bars = 200 μm, left). Quantification of the soft agar colony formation assay results (right). (B) Sphere formation assays revealing the self-renewal ability of ctrl and ORC1-overexpressing cells (scale bars = 200 μm, left). Quantification of the soft agar colony formation assay results (right). (C) Western-blotting assay indicating the regulations between AR and ORC1. (D) The scatter diagram indicating the positive correlations between AR and ORC1. (E) ChIP assays were utilized to determine the level of AR binding (left) and the

**Figure 7 (continued)**
enrichment of H3K27ac (right) at the promoter of ORC1 in AR deficiency or control PCa cells. (F) Cell proliferation determined in indicated cells with enzalumitide treatment. C4-2B-Parental and C4-2B-ENZR cells were transiently transfected as indicated. (G) Mice bearing C4-2B-ENZR xenografts were treated with contro DMSO and enzalutamide (10 mg/kg p.o.) for 4 weeks (*n* = 5/group). (H) The tumor volumes in indicated groups were detected the curves were generated. (I) The tumor weight in indicated groups were recorded and compared. ***$p < 0.001$.               

tumors and normal samples (*He et al., 2021*; *Sharp et al., 2019*). Nevertheless, bulk transcriptome profiling may ignore significant targets during the process. Thus, in the current study, we analyzed the scRNA-seq to identify top AR-related drivers and particularly screen the prognostic ARGs based on large PRAD cohorts. Then, we constructed the ARGs model integrating 10 genes to validate its predictive efficiency and robustness in internal and external cohorts. The ARGs score is an independent factor for PRAD progression and high levels of ARGs scores correlated with advanced stages, Gleason scores and tumor mutational burden. The consensus clustering analysis further classified the PRAD samples into two groups *via* ARGs gene set and found the highly enriched crosstalk in cluster 2 *versus* cluster 1. In addition, MTT assays were conducted to further narrow down the targets and we observed that ORC1 is a potential hit within these genes. Low-throughput validations revealed that targeting ORC1 could significantly suppress cell growth of PRAD cells. ORC1 expressed highly in PRAD samples relative to paired normal tissues and correlated with elevated Gleason scores. Kaplan-Meier analysis further suggested that ORC1 is a prognostic factor and patients with high ORC1 levels suffered from worse PFS outcomes. Mechanistically, AR could directly bind at the promoter of ORC1 that contributes to its enhanced transcription. Increased ORC1 levels could notably mediate the Enza-R process, whereas targeting ORC1 could render PRAD more sensitive to enzalutamide treatment.

As is well documented, aberrant activation of AR signaling could drive the progression and ADT resistance of PRAD, which is the main reason that contributes to limited efficacy of treatment (*Wen, Niu & Huang, 2020*; *VanDeusen et al., 2020*). However, there is currently no systematical screening of AR-related drivers in PRAD. Meanwhile, previous studies have already established the corresponding risk models based on autophagy-related signature, long non-coding RNAs, as well as DEGs (*Hu et al., 2020*, *2021*). Compared with these common models, the predictive efficiency and accuracy of ARGs are superior to most of these already established signature. In our model, we also analyzed the prognostic significance of well-known PRAD risk factors, including PSA levels and Gleason scores. Compared with other clinical factors, we found that ARGs signature and Gleason scores were all independent elements for PRAD PFS and prognosis, but not the T or N stages. Meanwhile, DCA indicated that incorporation of ARGs and Gleason scores could notably elevate the net benefit for PRAD patients. How to integrate the ARGs and Gleason scores together to enhance predictive significance remains an intriguing problem to elucidate. In addition, we also utilized the cluster analysis to stratify PRAD samples into two groups in which the patients had distinct outcomes. Cell cycle is the top hit that enriched in cluster 2 *versus* cluster 1, indicating the underlying relationships between ARGs gene set with cell

cycle process. Intriguingly, within these 10 identified genes, ORC1, ORC1, CENPF and DYNLL1 were already indicated to participate in cell cycle. For instance, one study found that DEFA1B could interact with ORC1 which is indispensable to initiate DNA replication during the cell cycle and enhanced DEFA1B proteins could inhibit the ORC1 level in the cell nuclei. *Li et al. (2021)* also indicated that overexpression of CENPF could accelerate cell proliferation and cell cycle, which correlated with progression and poor prognosis of lung adenocarcinoma. Collectively, we speculated that activated cell cycle process linked the tight associations between ARGs and PRAD progression.

The initial step of DNA replication contains the ORC in eukaryotes, consisting of ORC1/2/3/4/5/6 (*Lee et al., 2021*). As the largest subunit, ORC1 was originally recognized to be a component of DNA replication initiation complexes (*Kuo et al., 2012*). Besides, ORC1 could also function as a transcriptional regulator of a list of target genes, including human aldolase B and c-Myc genes (*Eladl et al., 2021*). Defective mutations of several ORC subunits in multicellular organisms would lead to pleiotropic phenotypes, like cell cycle arrest, zygotic lethality, as well as chromosomal abnormalities (*Chou et al., 2021*). Previous studies have indicated that ORC1 levels were regulated during the cell division cycle, and ORC is a dynamic complex (*Li et al., 2020*). ORC1 is ubiquitinated for destruction in the S phase entry, followed by the dissociation of ORC from chromosomes. While, ORC1 re-localizes to condensing chromatin during early mitosis and then exhibits various nuclear localization patterns during G1 phase, as evidenced by the time lapse and live cell images of human cells expressing fluorescently tagged ORC1 (*Higa et al., 2021*). *Xiong et al. (2020)* indicated that ORC1 played important roles in the migration, invasion, apoptosis, and proliferation of glioma cells. Mechanistically, OCR1 inhibition could suppress the activation of the Extracellular Signal-Regulated Kinase/c-Jun N-terminal Kinase (ERK/JNK) signaling pathway, implicating that ORC1 could be a novel prognostic marker of glioma *via* the activation of the ERK/JNK signaling pathway (*Xiong et al., 2020*). Besides, TRF2 is regarded to play an essential role in ORC and MCM loading at telomeres, and TRF2 directly binds to ORC complex *via* the ORC1 protein. The TRF2-ORC1 interaction thus promote the replication origins that are assembled at telomeres. The initiation events could have an important role in telomere maintenance because the persistent arrest of replication forks within a telomere would cause under replication for the absence of a converging fork. However, little research has elucidated the specific mechanisms that cause high levels of ORC1 in tumors. In this study, we utilized the ChIP-qPCR assay to find that AR could cooperate with H3K27ac, one active indicator of transcription, to promote the transcriptional activity of ORC1. ORC1 is the direct downstream target of AR in PRAD cells, which has never been reported. In line with the above knowledge, we found that ORC1 overexpression could significantly enhance cell stemness potentiality, as indicated by the sphere formation assays. Given that sustained activation of cell cycle and stemness features could contribute to Enza-R, we further demonstrated that ORC1 could mediate Enza-R based on the *in vitro* and *in vivo* assays.

However, this study still has several shortcomings that need to be further improved. First of all, we should have recruited more PRAD samples and cohorts to validate the predictive efficiency and robustness of ARGs model. Secondly, apart from ORC1, we

observed that targeting other genes in ARGs signature could also reduce cell growth. Experimental assays were further warranted to validate the functional roles of other targets, such as ORC or SHANK3. Lastly, although we have elucidated the relationships between AR-activated ORC1 levels with Enza-R, the underlying mechanisms were still uncertain. Whether ORC1 regulates other unknown regulators to drive Enza-R process would provide more useful vulnerabilities for treatment. Previous studies also indicated that stabilizing androgen receptor in mitosis could inhibits prostate cancer progression, indicating that we should choose suitable conditions to interfere ARG signaling (*Vander Griend, Litvinov & Isaacs, 2007*).

In summary, our study integrated the scRNA-seq and TCGA-PRAD data to identify a prognostic ARGs model for PRAD patients. The established ARGs score is a vital factor in PRAD that correlated with Gleason scores, tumor stages and progression-free survival outcomes. Experimental studies further screened that AR-activated ORC1 is an essential hit and could promote cell growth and stemness features. Targeting ORC1 renders PRAD cells more sensitive to enzalutamide therapy, which provides a valuable vulnerability that deserves further investigations.

## ABBREVIATIONS

| | |
|---|---|
| **PRAD** | prostate adenocarcinoma |
| **Enza-R** | enzalutamide resistance |
| **ARGs** | AR-associated genes |
| **ADT** | androgen deprivation therapy |
| **GSEA** | gene set enrichment analysis |
| **LASSO** | least absolute shrinkage and selection operator |
| **ATCC** | American Type Culture Collection |
| **DEGs** | differentially expressed genes |
| **DCA** | decision curve analysis |
| **AUC** | area under the curve |
| **CRPC** | castration-resistant PCa |

### Funding

The project was supported by the Basic Research Programs of Science and Technology Department of Wenzhou (Y20210171) and the Wenzhou associated of the Integration of Traditional and Western Medicine, clinical research fund (2021004). The funders had no role in study design, data collection and analysis, decision to publish, or preparation of the manuscript.

## Grant Disclosures

The following grant information was disclosed by the authors:
Science and Technology Department of Wenzhou: Y20210171.
Integration of Traditional and Western Medicine, Clinical Research: 2021004.

## Competing Interests

All authors declare that they have no competing interests.

## Author Contributions

- Linjin Li conceived and designed the experiments, performed the experiments, analyzed the data, authored or reviewed drafts of the article, and approved the final draft.
- Dake Chen conceived and designed the experiments, prepared figures and/or tables, and approved the final draft.
- Xiang Chen performed the experiments, authored or reviewed drafts of the article, and approved the final draft.
- Jianlong Zhu analyzed the data, authored or reviewed drafts of the article, and approved the final draft.
- Wenshuo Bao performed the experiments, prepared figures and/or tables, and approved the final draft.
- Chengpeng Li performed the experiments, prepared figures and/or tables, and approved the final draft.
- Feilong Miao performed the experiments, authored or reviewed drafts of the article, and approved the final draft.
- Rui Feng conceived and designed the experiments, performed the experiments, analyzed the data, prepared figures and/or tables, authored or reviewed drafts of the article, and approved the final draft.

## Animal Ethics

The following information was supplied relating to ethical approvals (*i.e.*, approving body and any reference numbers):

The Institutional Animal Experimental Ethics Committee of the Third Clinical Institute Affiliated to Wenzhou Medical University (Approval no. WMU2071357-AC-03).

## Data Availability

All original figures are available in the Supplemental Files.
GSE99795; GSE116918; GSE70769, MSKCC-PRAD cohort.

## Supplemental Information

Supplemental information for this article can be found online at http://dx.doi.org/10.7717/peerj.16850#supplemental-information.

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
