# Peer review of "An androgen receptor-based signature to predict prognosis and identification of ORC1 as a therapeutical target for prostate adenocarcinoma"

_PeerJ, doi:10.7717/peerj.16850_

## Round 0.1 · original submission · Major Revisions

It requires a number of Major Revisions.

Reviewer 1 ·

Basic reporting

In this study, Li et al demonstrated that ORC1 is one of the hub androgen receptor-associated genes that promotes proliferation and stem-like properties of PCa cells. This manuscript is quite interesting. Overall, the English is good with little typos to be revised.

Experimental design

The study design is scientific and the data is sufficient to address the current concern.

Validity of the findings

Some problems need to be corrected before publication.
1.The paper requires editing for language clarity. There are many typing and grammatical mistakes, and the language structure needs to by optimized.
2.shRNA sequences of ORC1 need to be clarified.
3.What does “147 samples of GSE99795” in line 115 refer to?
4.The rationale for selecting ORC1 for further analysis is a little bit confusing. Do you mean that the silencing of ORC1 led to the greatest effect on suppressing cell growth? How many candidates were involved for knockdown assay?
5.shRNAs were used for ORC1 knockdown in the M&M section, however, in the results section, a siRNA is mentioned.

Reviewer 2 ·

Basic reporting

The manuscript entitled "An AR-based signature to predict prognosis and identification of ORC1 as a therapeutical target in prostate adenocarcinoma" is of interesting to the scientific community however, mostly because of not clear writing the manuscript is confusing, and some important information is lost.

Experimental design

no comment

Validity of the findings

no comment

Additional comments

1. There are several typos, grammatical errors, and confusing statements in places, which reduce the readability of the text. Meticulous editing is required.
2. Introduction, “According to……34,130 in 2021”, a regional scope for this statistic data is required.
3. Please give the definitions of all abbreviated terms at their first presence in the man text.
4. For animal experiments, more detailed information of animals is required. Such as body weight and provider.
5.“We were not blinded for animal experiment” in line 216, why?.
6. The formula for xenograft tumor size calculation needs to be clarified.
7. Were the mice treated with enzalutamide? If so, what’s the detailed treatment strategy?

---

## Round 0.2 · Minor Revisions

This subject has been covered in previous articles, and it appears that some critical references have been omitted:

- Donald J. Vander Griend, Ivan V. Litvinov & John T. Isaacs (2007) Stabilizing Androgen Receptor in Mitosis Inhibits Prostate Cancer Proliferation, Cell Cycle, 6:6, 647-651,
DOI: 10.4161/cc.6.6.4028
- https://doi.org/10.1016/j.ccell.2021.03.010
- https://doi.org/10.1016/j.drudis.2021.01.034

There may be more, please do a thorough literature search.

Reviewer 1 ·

Basic reporting

No comment

Experimental design

No comment.

Validity of the findings

No comment

Reviewer 2 ·

Basic reporting

no comment

Experimental design

no comment

Validity of the findings

no comment

Additional comments

no comment

---

## Round 0.3 · accepted · Accept

The authors have addressed all of the reviewers' comments. The manuscript is accepted.